# Extracellular Vesicles Containing MicroRNA-92a-3p Facilitate Partial Endothelial-Mesenchymal Transition and Angiogenesis in Endothelial Cells

**DOI:** 10.3390/ijms20184406

**Published:** 2019-09-07

**Authors:** Nami O. Yamada, Kazuki Heishima, Yukihiro Akao, Takao Senda

**Affiliations:** 1Department of Anatomy, Graduate School of Medicine, Gifu University, 1-1 Yanagido, Gifu 501-1194, Japan; 2United Graduate School of Drug Discovery and Medical Information Sciences, Gifu University, 1-1 Yanagido, Gifu 501-1194, Japan

**Keywords:** extracellular vesicle, miR-92a-3p, angiogenesis, claudin-11

## Abstract

Extracellular vesicles (EVs) are nanometer-sized membranous vesicles used for primitive cell-to-cell communication. We previously reported that colon cancer-derived EVs contain abundant miR-92a-3p and have a pro-angiogenic function. We previously identified Dickkopf-3 (Dkk-3) as a direct target of miR-92a-3p; however, the pro-angiogenic function of miR-92a-3p cannot only be attributed to downregulation of Dkk-3. Therefore, the complete molecular mechanism by which miR-92a-3p exerts pro-angiogenic effects is still unclear. Here, we comprehensively analyzed the gene sets affected by ectopic expression of miR-92a-3p in endothelial cells to elucidate processes underlying EV-induced angiogenesis. We found that the ectopic expression of miR-92a-3p upregulated cell cycle- and mitosis-related gene expression and downregulated adhesion-related gene expression in endothelial cells. We also identified a novel target gene of miR-92a-3p, claudin-11. Claudin-11 belongs to the claudin gene family, which encodes essential components expressed at tight junctions (TJs). Disruption of TJs with a concomitant loss of claudin expression is a significant event in the process of epithelial-to-mesenchymal transition. Our findings have unveiled a new EV-mediated mechanism for tumor angiogenesis through the induction of partial endothelial-to-mesenchymal transition in endothelial cells.

## 1. Introduction

MicroRNAs (miRNAs) are small noncoding RNAs that inhibit gene expression in a posttranscriptional manner. The mRNAs complementarily matched to the seed position (nucleotides 2–8) of miRNAs have the potential to be under miRNA regulation. Therefore, in theory, a single miRNA strand has the potential to regulate more than 200 target mRNAs. Today, over 2000 miRNAs have been identified in the human genome and more than 30% of our gene sets are predicted to be conserved targets for miRNA regulation [1]. A large body of evidence has documented that miRNAs play a critical role in various biological processes, including proliferation, differentiation, and apoptosis. While the majority of miRNAs exist and work within cells, recent studies have shown that miRNAs also exist in the bodily fluids packed within extracellular vesicles (EVs), which are membranous vesicular structures derived from endosomes or the plasma membrane. These miRNAs are called circulating miRNAs and are considered novel mediators of intercellular communication. Indeed, recent accumulating evidence suggests that cancer cells secrete large amounts of EVs, transmit functional EV cargo to recipient cells, and manipulate the tumor microenvironment. EVs and their cargo are now highlighted as significant players in cancer development, progression, and metastasis. Among them, miR-92a-3p has been identified as a miRNA present in the plasma that is significantly upregulated in colon cancer patients compared with healthy individuals, suggesting that miR-92a-3p could be a potential noninvasive biomarker for colon cancer [2]. In our previous study, we showed that miR-92a-3p expression in the plasma is significantly downregulated after colon tumor resection [3]. Furthermore, we demonstrated that the EVs secreted by colon cancer are enriched with miR-92a-3p and promote angiogenesis, as indicated by increased proliferation, motility, and tube formation in human umbilical vein endothelial cells (HUVECs) through the inhibition of Dickkopf-3 (Dkk-3) expression [4]. However, this observation just described a function of EVs and/or miR-92a-3p and the detailed molecular mechanism by which EVs and/or miR-92a-3p exerts pro-angiogenic effects is still unclear. Therefore, in this study, we performed high-throughput analysis of gene sets affected by the ectopic expression of miR-92a-3p to understand, as a whole, the processes occurring in endothelial cells that incorporate EVs. Our findings suggest a new EV-based mechanism underlying how colon cancer influences surrounding cells and progresses.

## 2. Results 

### 2.1. Colon Cancer Cells Actively Secrete MiR-92a-3p via EVs

Previously, we demonstrated that miR-92a-3p is significantly upregulated in colon carcinoma tissues compared with colon adenoma tissues [4]. Furthermore, a significantly abundant amount of miR-92a-3p was found to exist in the plasma of colon cancer patients and xenografted mice, suggesting specific functions in the extracellular space of colon cancer. As shown in Figure 1A, miR-92a-3p expression was significantly upregulated in colon cancer cell lines compared with normal colon mucosa. Furthermore, miR-92a-3p levels in the extracellular space (packed within the EVs) were significantly higher than those within the cells (Figure 1B). These results indicate that colon cancer cells actively release miR-92a-3p via EVs. To further characterize the EVs derived from DLD-1 cells, the protein expression profiles of EVs were examined. CD63, CD81, and TSG101 are often used as identification markers for exosomes. Flotillin-1, a lipid raft-associated membranous protein, and actinin-4, a cytoskeletal protein, are used as markers for shed-microvesicles [5]. Western blot analysis demonstrated that CD63, CD81, and TSG101 were predominantly expressed in EVs compared with their paired cells (Figure 1D). As shown in Figure 1C, nanoparticle tracking analysis (NTA) revealed that the average size of DLD-1 cell-derived EVs was 82.8 ± 36.4 nm. Transmission electron microscopy (TEM) was also used to visualize the membranous vesicular morphology of EVs, and their sizes were found to be mostly smaller than 150 nm in diameter (Figure 1E). All these results indicate that the EVs collected from DLD-1 cells are mainly exosomes (30–150 nm).

### 2.2. EVs and MiR-92a-3p Promote Proliferation, Migration, and Tube Formation in HUVECs

Manipulation of a recipient cell’s phenotype can be achieved depending on the efficiency of EV uptake and transfer of its cargo. According to this theory, we first confirmed the presence of EV uptake by incubating HUVECs with fluorescence-labeled EVs derived from DLD-1 cells. As shown in Figure 2A, the labeled EVs were visualized as green dot-like shapes. The largest number of green dots was observed 16 h after the incubation started. The green dots were mainly localized around the nuclei of the HUVECs. After the 16 h time point, the number of green dots gradually decreased until 24 h post incubation (data not shown). This observation indicates that DLD-1 cell-derived EVs can be efficiently incorporated into HUVECs. Furthermore, the incorporated EVs significantly promoted the proliferation of HUVECs (Figure 2B) and also increased the intracellular levels of miR-92a-3p in HUVECs (Figure 2C). Ectopic expression of miR-92a-3p in HUVECs produced the same results (Figure 2D,E). Additionally, the incorporated EVs significantly promoted migration and tube formation in HUVECs (Figure 2F,G). Ectopic expression of miR-92a-3p in HUVECs also reproduced those results (Figure 2H,I), as previously demonstrated [4]. These findings indicate that the EVs containing miR-92a-3p secreted by colon cancer cells induce a pro-angiogenic phenotype in endothelial cells.

### 2.3. MiR-92a-3p Upregulates Cell-Cycle- and Mitosis-Related Genes and Downregulates Adhesion-Related Genes in HUVECs

After confirmation of the phenomena mentioned above, cDNA microarray analysis of the HUVECs transfected with miR-92a-3p versus those transfected with nonspecific control miRNA was performed. Figure 3A is a heatmap showing the expression patterns of 1232 differentially expressed genes (DEGs) in the two conditions (HUVECs transfected with miR-92a-3p or nonspecific control miRNA). Among them, we observed more than a four-fold upregulation or downregulation of DEGs by miR-92a-3p compared with the control, as summarized in Table 1 and Table 2. As shown in Table 1, the expression of mitosis- and cell-cycle-related genes was upregulated, including that of *KIF15* (Gene Ontology (GO):0007018 microtubule-based movement, GO:0007067 mitosis, and GO:0008283 cell proliferation) and *DSCC1* (GO:0006260 DNA replication, GO:0007049 cell cycle, and GO:0034088 maintenance of mitotic sister chromatid cohesion). Signaling through G protein-coupled receptors, which are involved in cancer biology including vascular remolding, invasion, and migration, was also activated, as indicated by the upregulation of *RGS7* (GO:0007186 G-protein coupled receptor protein signaling pathway and GO:0007242 intracellular signaling cascade) and *C14orf174* (GO:0007186 G-protein coupled receptor protein signaling pathway). As shown in Table 2, the target genes of miR-92a-3p that have previously been validated, including *Dkk-3* [4,6], *CD69* [7], and *Integrin subunit alpha 5* (*ITGA5*) [8], were present in the four-fold downregulated gene list. The results of the modular enrichment analysis of the upregulated or downregulated DEGs by miR-92a-3p are summarized in Figure 3B. The upregulated DEGs were enriched in “cell cycle”, “mitosis”, and “cell division, mitosis”. The downregulated DEGs were enriched in “signal transduction”, “immune response”, “cytokine-mediated signaling pathway”, “cell adhesion”, and “proteolysis”.

### 2.4. CLDN11 is a Novel Target Gene of MiR-92a-3p


*CLDN11* belongs to the claudin gene family, which encodes integral components of tight junctions (TJs). In normal epithelia, TJs have various functions, including the regulation of paracellular transport, cell–cell adhesion, cell polarity, proliferation, and differentiation. The disruption of TJs with a loss of TJ-related proteins is a critical process in cancer development and metastasis. As shown in Table 2, *CLDN11* is included in the list of DEGs that showed a four-fold downregulation by miR-92a-3p and is also a potential target gene of miR-92a-3p, as predicted by the TargetScanHuman 7.1 (http://www.targetscan.org/vert_71/). We first confirmed that the expression levels of *CLDN11* mRNA in the four colon cancer cell lines were significantly downregulated compared with those in normal colon tissue (Figure 4A). The incorporated EVs and the ectopic expression of miR-92a-3p significantly downregulated CLDN11 mRNA and protein in HUVECs (Figure 4B,C). To confirm *CLDN11* as a target gene of miR-92a-3p, we cloned the 3’UTR of *CLDN11* mRNA, the sequence containing the predicted binding site of miR-92a-3p (region A), into the pMIR-REPORT vector (pMIR-A; Figure 4D). The luciferase activity of wild-type pMIR-A was significantly decreased when the cells were incubated with EVs or co-transfected with miR-92a-3p, whereas mutation of the seed sequence abolished the ability of EVs and miR-92a-3p to inhibit luciferase expression (Figure 4D). These results indicate that miR-92a-3p directly regulates *CLDN11* expression in HUVECs. Gene silencing of *CLDN11* (Figure 4F) also promoted proliferation (Figure 4E), migration (Figure 4G), and tube formation (Figure 4H) in HUVECs. These results indicate that CLDN11 downregulation by EVs containing miR-92a-3p also contributes to the pro-angiogenic state of endothelial cells.

### 2.5. MiR-92a-3p Induces Partial Endothelial-to-Mesenchymal Transition (EndoMT) in HUVECs

The biological process-based functional network of miR-92-3p target genes (*CLDN11*, *Dkk-3*, *CD69*, and *ITGA5*) was depicted using GeneMANIA (https://genemania.org/; Figure 5A). The network contains TJ-related genes (*CLDN11*, *PATJ*, *TJP3*, and *TJP1 (ZO-1)*), extracellular matrix/structure organization-related genes (*ITGA5*, *SPP1*, *ITGB1*, *ITGB3*, *COL18A1*, and *FBN1*), vascular endothelial growth factor receptor signaling pathway-related genes (*ITGA5*, *ITGB3*, and *VEGFD*), and cell–cell junction-related genes (*CLDN11*, *ITGA5*, *ITGB1*, *PATJ*, *TJP3*, *TJP1 (ZO-1)*, and *CD9*). All these genes have the potential to be regulated by miR-92a-3p directly or indirectly. As many TJ-related and cell adhesion-related genes are regulated by miR-92a-3p, we hypothesized that EVs containing miR-92a-3p induce phenomena similar to the epithelial-to-mesenchymal transition (EMT) in HUVECs. Thus, we further explored EMT-related signaling in HUVECs incubated with EVs or transfected with miR-92a-3p. Transfection of DLD-1 cells with antagomiR-92a-3p significantly decreased the intracellular level of miR-92a-3p and miR-92a-3p secretion via EVs (Figure 5B). Furthermore, incubation with EVs or ectopic expression of miR-92a-3p upregulated snail and vimentin—mesenchymal markers—in HUVECs (Figure 5C,D) and downregulated ZO-1, an epithelial marker; however, the downregulation of ZO-1 was not as drastic (Figure 5C). EVs isolated from DLD-1 cells transfected with antagomiR-92a-3p impaired the EV-mediated upregulation of snail and vimentin (Figure 5C,D). These results suggest that miR-92a-3p is involved in the induction of an EMT-like process called partial EndoMT.

## 3. Discussion

The present study aimed to determine why colon cancer cells secrete miR-92a-3p via EVs into the surrounding environment. We found that colon cancer cells secrete miR-92a-3p via EVs to exert the following effects in endothelial cells: accelerating the cell cycle, including mitosis, which leads to endothelial cell proliferation;loosening intercellular adhesions, which promotes migration;promoting tube formation through direct regulation of miR-92a-3p target genes, including a newly identified target gene (*CLDN11*), and indirectly regulating their related genes.

We conclude that, as a whole, these pro-angiogenic processes can contribute to the induction of partial EndoMT in endothelial cells. Consequently, cancer cells receive access to the host’s vessels, which may be utilized as a feeding and/or metastasis route; this supposition should be validated in future research. 

We first confirmed that EVs containing miR-92a-3p were efficiently incorporated into endothelial cells, consequently promoting proliferation, migration, and tube formation in endothelial cells. These phenomena were also observed in our previous study, in which Dkk-3 was identified as a target gene of miR-92a-3p [4]. However, the discovery of the regulation of Dkk-3 by miR-92a-3p could not fully explain the complete function of EVs and/or miR-92a-3p in endothelial cells. Thus, cDNA microarray analysis greatly contributed to further revealing the molecular mechanisms underlying these pro-angiogenic events. Heatmap and clustering analysis revealed gene sets that were significantly upregulated or downregulated by miR-92a-3p transfection in endothelial cells. According to the results of the GO analyses, miR-92a-3p significantly upregulated “cell-cycle”, “mitosis”, and “cell division”-related genes, suggesting that the increased cell proliferation of the HUVECs observed in this study can likely be attributed to those upregulated DEGs. GO analysis also demonstrated that miR-92a-3p downregulated “signal transduction”, “immune response”, “cytokine-mediated signaling pathway”, “cell adhesion”, and “proteolysis”-related gene expression. Among them, I*TGA5*, *Dkk-3*, and *CD69*—already reported target genes of miR-92a-3p—were listed as significantly downregulated genes. We also identified a novel target gene of miR-92a-3p—*CLDN11*.

Claudins, integral membrane proteins in TJs, are expressed in all epithelia and endothelia; they determine the permeability of TJs via modulation of paracellular pathways. So far, 26 claudins that are expressed in a tissue-specific manner have been identified in humans. Among them, CLDN11 was initially identified as oligodendrocyte-specific protein (OSP), as it is highly expressed by oligodendrocytes of the central nervous system (CNS) and localized in the interlamellar TJ strands of myelin sheaths [9,10]. CLDN11/OSP is also strongly expressed at TJs between Sertoli cells of the testicles, forming the blood–testis barrier [11]. Accordingly, CLDN11/OSP knockout mice exhibit both neurological and reproductive disorders, including slow nerve conduction velocity and infertility in male mice due to deficits of the parallel-array TJ strands in the CNS myelin and Sertoli cells [12]. Subsequently, claudin-11/OSP expression in endothelial cells and its barrier function has also been demonstrated [13,14]. Recent studies have shown that the downregulation of CLDN11/OSP contributes to the malignancy of various cancers, as indicated by an increased degree of invasiveness, metastasis promotion, and poor prognoses in colon cancer [15], gastric cancer [16], and laryngeal squamous cell carcinoma [17] while the forced expression of CLDN11/OSP in bladder cancer decreases the degree of invasiveness [18]. Those studies demonstrated that the decreased expression level of CLDN11/OSP in cancer is due to the hypermethylation of its promoter region. Our study showed that CLDN11/OSP is also regulated by miR-92a-3p in colon cancer. Altogether, loss of CLDN11/OSP, to which miR-92a-3p contributes, is considered a mechanism for the disruption of cell adhesion and increased cell motility, which are important processes not only in cancer progression and metastasis but also in angiogenesis.

The functional network of miR-92-3p target genes (*CLDN11*, *Dkk-3*, *CD69*, and *ITGA5*) drawn by GeneMANIA indicates that TJ-related genes, extracellular matrix organization-related genes, vascular endothelial growth factor receptor signaling pathway-related genes, and cell–cell junction-related genes have the potential to be under miR-92a-3p regulation directly or indirectly. This result also suggests that miR-92a-3p is involved in EndoMT. EndoMT is a variant of traditional epithelial-to-mesenchymal transition (EMT) in which endothelial cells lose their polarity and cell–cell contacts and acquire mesenchymal or myofibroblastic phenotypes, sharing some of the pathways and effectors with EMT. Although EndoMT was considered a rare event in the past, accumulating evidence suggests that endothelial cells undergo EndoMT to supply fibroblasts during many physiological and pathological processes, including embryonic vascular development [19], cardiac fibrosis [20], pulmonary hypertension [21], and cancer development [22,23,24]. The hallmarks of EndoMT are the upregulation of mesenchymal genes (vimentin and α-SMA) and downregulation of epithelial genes (ZO-1 and VE-Cadherin), manifesting in proliferative and migratory phenotypes. During the angiogenic process, endothelial cells at the tip of emerging sprouts are indeed migratory and without cell polarity but stay connected with their neighboring cells. The tip endothelial cells in the sprouts express mesenchymal genes; however, they retain epithelial gene expression. Owing to these characteristics, angiogenic endothelial cells are considered to be in a state of “partial” EndoMT [25,26]. We also confirmed that the HUVECs we used were proliferative and migratory in nature, with upregulated snail and vimentin expression. However, the degree of downregulation of ZO-1—an epithelial marker—was mild in our HUVECs, supporting the conclusion that EVs containing miR-92a-3p induce “partial” EndoMT.

In conclusion, we validated that colon cancer cell-derived EVs are enriched with miR-92a-3p and that EVs likely induced angiogenesis through the upregulation of cell-cycle/mitosis-related genes and downregulation of adhesion-related genes, manifesting in proliferative and migratory phenotypes in endothelial cells. The hallmarks of EMT, vimentin upregulation and mild ZO-1 downregulation, were also observed. Taken together, we conclude that EVs containing miR-92a-3p induce partial EndoMT in endothelial cells. EVs have strong potential to be used as therapeutic targets against cancer progression and metastasis. Further studies are needed to better understand the whole system of cell–cell communication that occurs via EVs so that EVs can be utilized in cancer therapy.

## 4. Materials and Methods

### 4.1. Cell Culture

Human colon cancer DLD-1 (Cat# JCRB9094), WiDr (Cat# JCRB0224), and COLO201 (Cat# JCRB0226) cells were obtained from the Japanese Collection of Research Bioresources (JCRB) Cell Bank (Osaka, Japan), while SW480 (SW-480) (ATCC^®^ CCL-228^TM^) cells were obtained from American Type Culture Collection (Manassas, VA). HUVECs were obtained from KURABO (Osaka, Japan). The colon cancer cells were maintained in RPMI-1640 (Sigma-Aldrich, St Louis, MO) containing 10% (*v*/*v*) heat-inactivated fetal bovine serum (FBS; Thermo Fisher Scientific, Waltham, MA), while the HUVECs were maintained in complete medium (Humedia-EG2; KURABO). All cell lines were cultured under an atmosphere of 95% air and 5% CO_2_ at 37 °C. The Trypan blue exclusion test was used to determine the number of viable cells in a cell suspension, from which the cell proliferation ratio was calculated.

### 4.2. Isolation of Colon Cancer Cell-Derived EVs

For the collection of colon cancer cell-derived EVs, RPMI-1640 medium supplemented with EV-deprived FBS was prepared as follows: FBS was centrifuged at 3000 rpm for 5 min; then its supernatant was filtered through a Millex-GP (PES) Filter Unit (0.22 µm pores). The flow-through fraction was ultracentrifuged at 100,000 rpm for 3 h (TLA-110 fixed-angle rotor, Optima TLX, Beckman Coulter, Fullerton, CA, USA). Without disturbing the EV pellet, the supernatant was carefully collected and used as EV-deprived FBS, which was then added to the RPMI-1640 medium (EV-deprived RPMI). DLD-1 and WiDr cells (1 × 10^6^ cells) were cultured for 96 h in flasks containing 40 mL of EV-deprived RPMI. The culture medium was then collected and centrifuged at 2000 rpm for 5 min. The resulting supernatant was filtered through a 0.22 µm pore filter for the removal of cell debris, large EVs, and/or apoptotic bodies. The flow-through fraction was ultracentrifuged at 100,000 rpm for 3 h. The EV pellet was resuspended in 1 mL of phosphate buffered saline (PBS). For the analysis of the miRNA expression profiles of the EVs, RNA lysis buffer (described in Section 4.6) was added to an EV pellet and RNA samples were prepared. To analyze the protein expression profile of EVs, protein lysis buffer (described in Section 4.9) was added to another EV pellet and protein samples were prepared. The EVs resuspended in PBS were added to the culture medium of HUVECs at a concentration of 0.5 × 10^11^–1 × 10^11^ particles/0.5 × 10^5^ cells.

### 4.3. Transmission Electron Microscopy (TEM)

EV samples were also observed by using TEM as follows: EVs resuspended in PBS were mounted on copper grids (200 mesh and coated by formvar carbon film) and stained with 2% phosphotungstic acid (PTA) at pH 7 for 2 min. Excess PTA on the grids was removed, and samples were dried. A JEOL 1010 TEM was used to observe EV samples at a voltage of 80 kV.

### 4.4. Nanoparticle Tracking Analysis (NTA)

NTA is a method used for the detection of secreted EVs in a liquid sample. Colon cancer cell-derived EVs suspended in PBS (described in Section 4.2) were analyzed using NTA Version 2.3 Build 0034 instrument (NanoSight, Wiltshire, UK). Samples were diluted at 1:1000 in PBS and analyzed.

### 4.5. Transfection with MiR-92a-3p, Antisense Inhibitor for MiR-92a-3p (AntagomiR-92a-3p), or Short-Interfering RNA for CLDN11

Cells were seeded into six-well plates at a concentration of 0.5 × 10^5^ per well (10–30% confluent) on the day before transfection. Mature-type miR-92a-3p (mirVana miRNA mimic; Ambion, Foster City, CA, USA), antagomiR-92a-3p (mirVana miRNA inhibitor; Ambion), or short-interfering RNA (siRNA) for *CLDN11* (siR-*CLDN11*) were transfected into the cells using Lipofectamine RNAiMAX (Invitrogen, Carlsbad, CA, USA) according to the manufacturer’s protocol. The sequence of the nonspecific control miRNA (Hokkaido System Science) was 5′-GUAGGAGUAGUGAAAGGCC-3′ [27]. The sequences of siR-CLDN11 were 5′-GACCACCAUCGUGAGCUUUUU-3′ and 5′-AAAGCUCACGAUGGUGGUCUU-3′. The effects manifested by the introduction of miR-92a-3p or siR-CLDN11 into the cells were assessed at 48 h after transfection.

### 4.6. RNA Isolation and Quantitative Real-Time PCR

Total RNA of human normal colon tissue was purchased from Clontech (Mountain View, CA, USA). Total RNA was isolated from cultured cells with RNAiso Plus reagent (TaKaRa, Otsu, Japan) followed by DNase I treatment. To evaluate mRNA expression, total RNA was reverse-transcribed with ReverTra Ace qPCR RT Master Mix (TOYOBO, Osaka, Japan). Quantitative real-time polymerase chain reaction (qRT-PCR) was then performed using THUNDERBIRD SYBR qPCR Mix (TOYOBO) and the following primers: *CLDN11* (sense, 5′-AATGACTGGGTGGTGACCTG-3′; antisense, 5′-CTGTACTTAGCCACACCGGG-3′) and *GAPDH* (sense, 5′-CCCAGAAGACTGTGGATGGC-3′; antisense, 5′-TGGGTGTCGCTGTTGAAGTC-3′). To determine the expression levels of miRNAs, we conducted qRT-PCR using TaqMan MicroRNA Assays (Applied Biosystems, Foster City, CA, USA) and THUNDERBIRD Probe qPCR Mix (TOYOBO) according to the manufacturer’s protocol. The relative expression levels were calculated by the ΔΔC_t_ method. *GAPDH* was used as an endogenous normalizer for mRNA expression. RNU6B was used as an endogenous normalizer for miR-92a-3p expression. MiR-21 was used as a common normalizer for intracellular and extracellular miR-92a-3p expression because we preliminarily confirmed the abundant presence of miR-21 in both the intracellular and extracellular compartments of the colon cancer cells.

### 4.7. DNA Microarray Analysis and Data Mining

HUVECs were transfected with nonspecific miRNA or miR-92a-3p (10 nM), and total RNA was extracted at 24 h after transfection. One microgram of total RNA in RNase-free water was submitted to Toray Industries (Tokyo, Japan). DNA microarray analysis of the HUVECs transfected with nonspecific miRNA versus those transfected with miR-92a-3p was performed using 3D-Gene Human oligo chip 25k ver. 2.10 (Toray Industries). The raw signals for each gene were normalized by the global normalization method (median of Cy3/Cy5 ratios = 1). DEGs were analyzed prior to the modular enrichment analysis. Genes with normalized Cy3/Cy5 ratios greater than 2.0 or lower than 0.5 were defined as significantly upregulated or downregulated DEGs by miR-92a-3p, respectively. The heatmaps for the DEGs were generated with GeneSpring 13.0 software (Agilent Technology, Santa Clara, CA, USA) using the clustering function and a Euclidean correlation as a distant metric. For the interpretation of the underlying biological processes, modular enrichment analysis was conducted using GeneCodis (http://genecodis.cnb.csic.es/). The microarray data have been deposited in the ArrayExpress database at EMBL-EBI (https://www.ebi.ac.uk/arrayexpress/) under accession number E-MTAB-8093.

### 4.8. Antibodies

The primary antibodies used in this study were as follows: anti-CD63 (dilution 1:1000, Santa Cruz Biotechnology, Santa Cruz, CA, USA), anti-CD81 (dilution 1:1000, MBL, Nagoya, Japan), anti-TSG101 (dilution 1:1000, Abcam, Cambridge, MA), anti-flotillin-1 (dilution 1:1000, Cell Signaling Technology, Danvers, MA), anti-actinin-4 (dilution 1:1000, Cell Signaling Technology), anti-β-actin (dilution 1:5000, Sigma-Aldrich), anti-CLDN11 (dilution 1:1000, R&D Systems, Minneapolis, MN), anti-vimentin (dilution 1:1000 in western blotting and 1:100 in immunocytochemistry, Cell Signaling Technology), anti-ZO-1 (dilution 1:1000, Cell Signaling Technology), and anti-Snail (dilution 1:1000, Cell Signaling Technology). The secondary antibodies used in this study were as follows: horseradish peroxidase (HRP)-conjugated horse anti-mouse IgG (dilution 1:1000, Cell Signaling Technology), HRP-conjugated goat anti-rabbit IgG (dilution 1:1000, Cell Signaling Technology), and Alexa Fluor 488-conjugated goat anti-rabbit IgG (dilution 1:1000, Thermo Fisher Scientific). 

### 4.9. Western Blotting

Harvested cells were lysed with ice-cold protein lysis buffer comprising 20 mM HEPES/NaOH, 1% Triton X-100, 1 mM ethylenediaminetetraacetic acid, 1 mM ethylene glycol-bis(β-aminoethyl ether)-N,N,N′,N′-tetraacetic acid, 150 mM NaCl, 0.5% sodium deoxycholate, 0.1% sodium dodecyl sulfate (SDS), protease inhibitor cocktail (aprotinin, leupeptin, phenylmethylsulfonyl fluoride, and pepstatin A), and phosphatase inhibitor cocktail set III. Protein concentrations were determined with a BCA Protein Assay Kit (TaKaRa). The protein lysates were subsequently mixed with SDS sample buffer and boiled, followed by separation (10 µg total) by sodium dodecyl sulfate–polyacrylamide gel electrophoresis (SDS-PAGE) using a 5–10% polyacrylamide gel. Proteins were electroblotted onto a polyvinylidene fluoride membrane (PerkinElmer, Boston, MA, USA), which was subsequently blocked with 5% nonfat dry milk in tris-buffered saline and polysorbate 20 (TBST) (137 mM NaCl, 20 mM Tris at pH 7.6, and 0.1% Tween 20) and incubated with primary antibodies overnight at 4 °C. The membrane was washed three times with TBST and incubated with the appropriate HRP-conjugated secondary antibody for 1 h at room temperature. The immunoblots were visualized using an enhanced chemiluminescent kit (Wako Pure Chemical Industries, Osaka, Japan) according to the manufacturer’s instructions. Densitometric values for each immunoblot were calculated using Image J.

### 4.10. Wound Healing Assay

HUVECs were seeded into 12-well plates at a concentration of 0.5 × 10^5^ per well two days before miR-92a-3p or siR-CLDN11 transfection. Wound healing assays were performed as described previously [4].

### 4.11. Tube Formation Assay

Tube formation assays were performed according to the manufacturer’s protocol of an Angiogenesis Starter Kit (Life Technologies, Carlsbad, CA, USA). HUVECs were seeded into six-well plates at a concentration of 0.5 × 10^5^ per well on the day before transfection. The assay was performed at 24 h after transfection.

### 4.12. Immunocytochemistry

HUVECs were seeded on eight-well chamber slides (Matsunami Glass Ind., Osaka, Japan) the day before incubation with DLD-1 cell-derived EVs or transfection with miR-92a-3p. The EVs were pre-stained with Vybrant DiO cell-labeling solution (Molecular Probes, Eugene, OR). Briefly, EVs in PBS were incubated with 5 µL of labeling solution for 20 min at room temperature. The labeled suspension was ultracentrifuged at 100,000 rpm for 3 h; then, the supernatant was removed. The labeled EV pellet was once again resuspended in PBS and ultracentrifuged at 100,000 rpm for 3 h as a wash procedure. The washed and labeled EVs were added to the culture media of HUVECs and incubated for 16 h. HUVECs incubated with the labeled EVs were then fixed with 4% paraformaldehyde in PBS for 15 min at room temperature, followed by coverslip mounting with VECTASHIELD Mounting Medium with 4′,6-diamidino-2-phenylindole (DAPI) (Vector Laboratories, Inc., Burlingame, CA, USA). HUVECs transfected with miR-92a-3p or nonspecific control miRNA were also fixed with 4% paraformaldehyde in PBS for 15 min at room temperature, incubated with anti-vimentin antibody and then secondary antibody, followed by coverslip mounting with the mounting medium containing DAPI. The stained cells were observed using Biorevo fluorescence microscopy BZ-X700 (Keyence, Osaka, Japan).

### 4.13. Luciferase Assay

HUVECs were seeded into 96-well plates at a concentration of 0.1 × 10^4^ per well on the day before transfection. On searching TargetScanHuman 7.1 (http://www.targetscan.org/vert_71/) to identify algorithm-based binding sites for miR-92a-3p, we found that the predicted binding site was present at position 2140–2147 in the 3′UTR of *CLDN11* mRNA. The sequence region 2052–2372, containing the putative binding sequence for miR-92a-3p, was inserted into a pMIR-REPORT Luciferase miRNA Expression Reporter Vector (Applied Biosystems) according to the manufacturer’s protocol. Moreover, we created another pMIR construct encompassing a mutated seed sequence for miR-92a-3p (wild type, GCAAUA; mutant, GUCGUA) using the Prime STAR Mutagenesis Basal Kit (TaKaRa). The mutation of the vector was confirmed by sequence analysis. A pRL-TK Renilla luciferase reporter vector (Promega, Madison, WI) was used as an internal control vector. HUVECs were co-transfected with each reporter vector (0.01 µg) and miR-92a-3p or nonspecific control miRNA (10 nM). Luciferase activity was measured at 24 h after co-transfection using a Dual-Glo Luciferase Assay System (Promega) according to the manufacturer’s protocol. Results were reported as the firefly luciferase/Renilla luciferase ratio.

### 4.14. Statistical Analysis

Where applicable, values were expressed as the mean ± standard deviation. Statistical significance was analyzed using two-tailed Student’s *t* tests. A *p* value lower than 0.05 was considered significant.

## Figures and Tables

**Figure 1 ijms-20-04406-f001:**
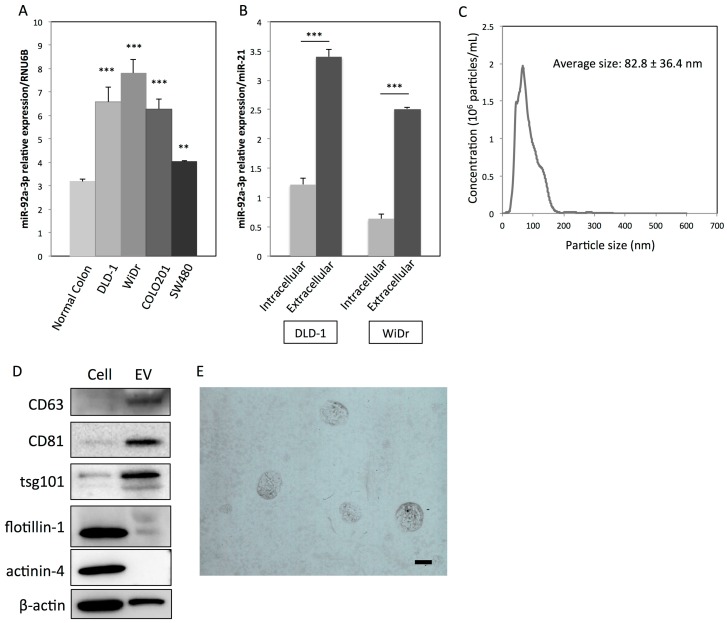
MiR-92a-3p is actively secreted via extracellular vesicles (EVs) from colon cancer cells. (**A**) Relative expression levels of miR-92-3p in four human colon cancer cell lines compared with those of normal colon mucosa: RNU6B was used as an endogenous normalizer for intracellular miR-92a-3p expression. (**B**) Relative expression levels of intra- and extracellular (within the EVs) miR-92a-3p in colon cancer DLD-1 and WiDr cells: MiR-21 was used as a common normalizer for intracellular and extracellular miR-92a-3p expression. ** *p* < 0.01; *** *p* < 0.001. (**C**) Characterization of EVs derived from DLD-1 cells by nanoparticle tracking analysis (NTA): The x-axis denotes particle size, while the y-axis denotes concentration (10^6^ particles/mL). (**D**) Protein expression profiles of EVs and corresponding DLD-1 cells: CD63, CD81, and TSG101 are markers for exosomes. Flotillin-1 and actinin-4 are markers for shed-microvesicles. (**E**) Characterization of EVs derived from DLD-1 cells by TEM. Scale bar: 100 nm. miRNA, microRNA; EV, extracellular vesicle.

**Figure 2 ijms-20-04406-f002:**
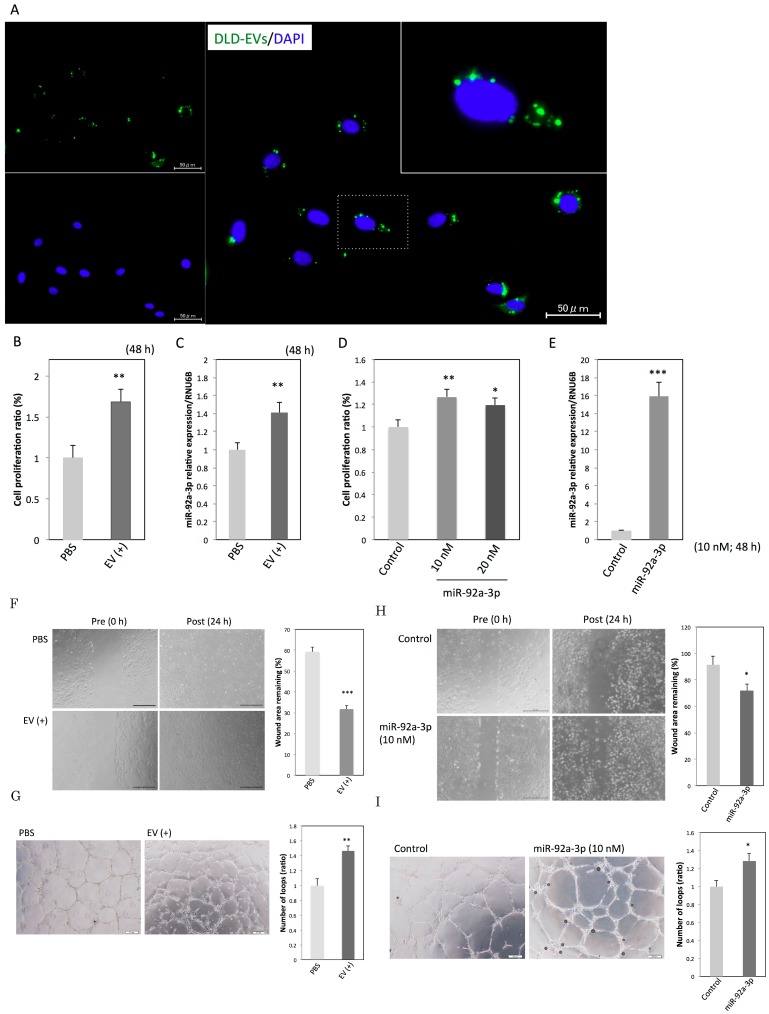
EVs enriched with miR-92a-3p induce a pro-angiogenic state in endothelial cells. (**A**) Immunostaining of human umbilical vein endothelial cells (HUVECs) at 16 h after the incubation with EVs was performed: EVs were derived from DLD-1 cells (green), and nuclei were from HUVECs (blue). Scale bars: 50 µm. The inset is an enlarged image of a nucleus and EVs. (**B**) Cell proliferation ratio and (**C**) relative expression levels of intracellular miR-92a-3p at 48 h after the incubation of HUVECs with PBS or EVs derived from DLD-1 cells. ** *p* < 0.01. (**D**) Cell proliferation ratio and (**E**) relative expression levels of intracellular miR-92a-3p at 48 h after the transfection of HUVECs with miR-92a-3p or nonspecific control miRNA. * *p* < 0.05, ** *p* < 0.01, and *** *p* < 0.001. (**F**) Migration assay and (**G**) tube formation assay in HUVECs incubated with PBS or EVs. Scale bars: 500 µm in Figure 2F and 200 µm in Figure 2G. ** *p* < 0.01; *** *p* < 0.001. (**H**) Migration assay and (**I**) tube formation assay in HUVECs transfected with miR-92a-3p or nonspecific control miRNA. Scale bars: 500 µm in Figure 2H and 200 µm in Figure 2I. * *p* < 0.05. miRNA, microRNA; EV, extracellular vesicle; HUVEC, human umbilical vein endothelial cell; PBS, phosphate buffered saline.

**Figure 3 ijms-20-04406-f003:**
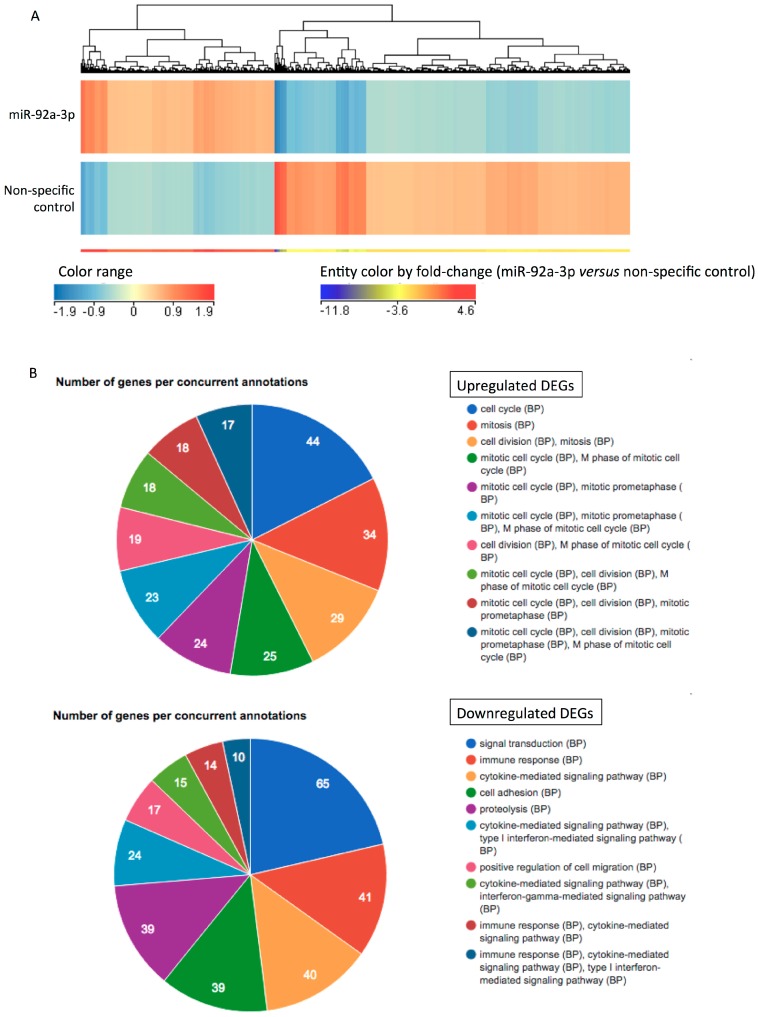
Heatmap and functional enrichment analysis reveal the occurrence of underlying molecular events in HUVECs transfected with miR-92a-3p. (**A**) Heatmap and hierarchical clustering analysis of differentially expressed genes (DEGs; *p*-value < 0.05 and fold-change ≥ 2) in HUVECs transfected with miR-92a-3p compared with those in HUVECs transfected with nonspecific control miRNA. Each column represents the relative expression level of individual genes. The red or blue color indicates relatively high or low expressions, respectively. Entity colors by fold-change (miR-92a-3p/nonspecific control) are also described below the heatmap. (**B**) Functional enrichment analysis of up- and downregulated DEGs, including the top 10 gene ontology terms. BP, biological process; miRNA, microRNA; EV, extracellular vesicle; HUVEC, human umbilical vein endothelial cell; DEG, differentially expressed gene.

**Figure 4 ijms-20-04406-f004:**
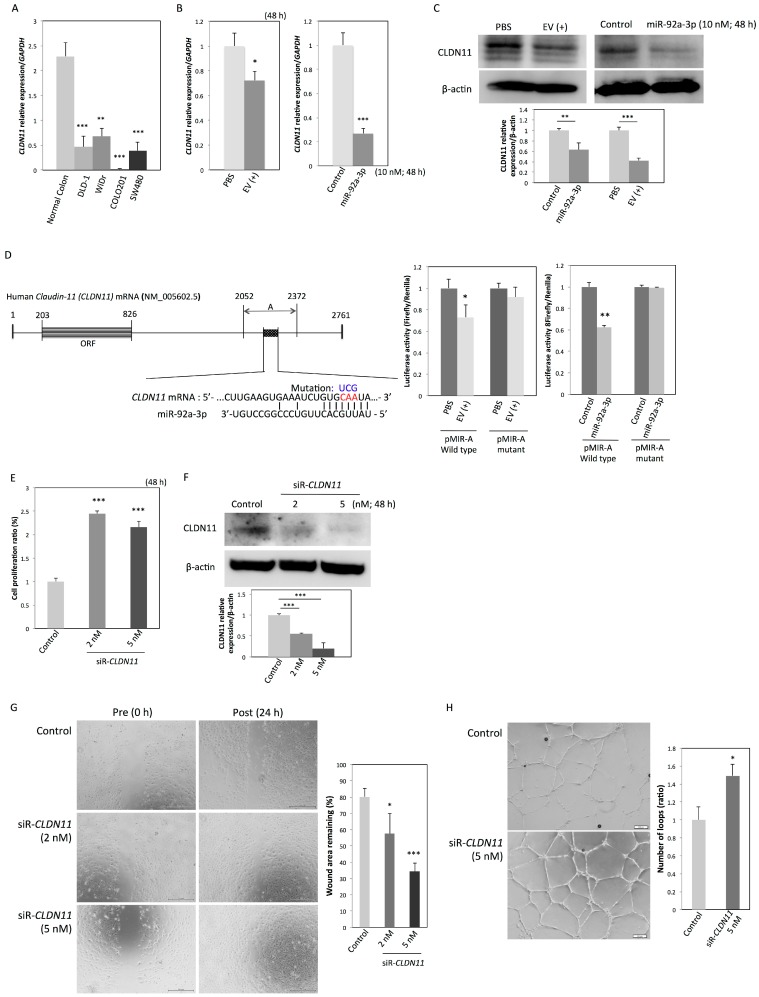
MiR-92a-3p induces a pro-angiogenic state in endothelial cells through the downregulation of *CLDN11*. (**A**) Relative expression levels of *CLDN11* mRNA in four colon cancer cell lines: *GAPDH* was used as an endogenous normalizer for *CLDN11* expression. ** *p* < 0.01; *** *p* < 0.001. (**B**) Relative expression levels of *CLDN11* mRNA and (**C**) CLDN11 protein in HUVECs at 48 h after incubation with EVs or transfection with miR-92a-3p or nonspecific control miRNA. * *p* < 0.05, ** *p* < 0.01, and *** *p* < 0.001. (**D**) Predicted binding site for miR-92a-3p in the 3’UTR region of human *CLDN11* mRNA (region A shown as a dotted box). Mutant-type pMIR contained a mutated seed sequence (from CAA to UCG). Luciferase activities after incubation with EVs or co-transfection of HUVECs with miR-92a-3p or nonspecific control miRNA and wild-type or mutant pMIR-A vector. ** *p* < 0.01. (**E**) Cell proliferation ratio and (**F**) relative expression levels of CLDN11 protein at 48 h after transfection of HUVECs with siR-*CLDN11* or nonspecific control miRNA. *** *p* < 0.001. (**G**) Migration assay and (**H**) tube formation assay in HUVECs transfected with siR-*CLDN11* or nonspecific control miRNA. Scale bars: 500 µm in Figure 4G and 200 µm in Figure 4H. * *p* < 0.05; **** p* < 0.001. miRNA, microRNA; HUVEC, human umbilical vein endothelial cell.

**Figure 5 ijms-20-04406-f005:**
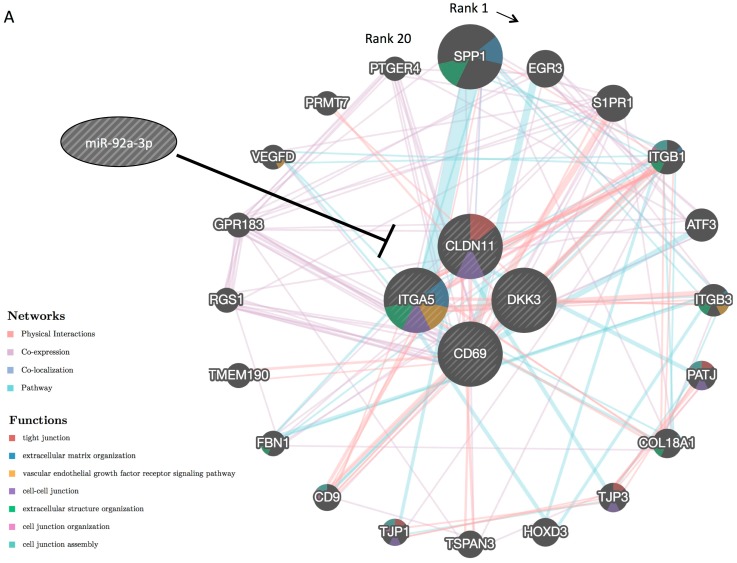
MiR-92a-3p induces “partial” Endothelial-to-Mesenchymal Transition (EndoMT; a hypothesis). (**A**) Functional interaction network constructed from miR-92-3p target genes (*CLDN11*, *Dkk-3*, *CD69*, and *ITGA5*) using the GeneMANIA algorithm (https://genemania.org/): Network weighting is based on biological processes. Genes are arranged clockwise from high to low in terms of rank. A higher rank denotes that a gene is likelier to be related to the miR-92a-3p target genes arranged in the center. Networks are connected by physical interactions (73.97%), co-expression (14.65%), co-localization (6.61%), and pathway (4.76%). (**B**) Intracellular and extracellular levels of miR-92a-3p after the transfection of DLD-1 cells with nonspecific control miRNA or antagomiR-92a-3p at 20 nM. (C-EV, EVs isolated from nonspecific control miRNA-transfected DLD-1 cells; A-EV, EVs isolated from antagomiR-92a-3p-transfected DLD-1 cells. EVs were isolated at 96 h after the transfection). (**C**) Protein expression profiles of HUVECs incubated with EVs or transfected with miR-92a-3p. (C-EV(+), HUVECs incubated with C-EV for 48 h; A-EV(+), HUVECs incubated with A-EV for 48 h). Vimentin is a mesenchymal marker whereas ZO-1 is an epithelial marker. N.S., not significant; * *p* < 0.05, ** *p* < 0.01, and *** *p* < 0.001. (**D**) HUVECs immunostained at 48 h after incubation with EVs or transfection: Vimentin (green), nuclei of the HUVECs (blue). Scale bars: 50 µm. EndoMT, endothelial-mesenchymal transition; miRNA, microRNA; HUVEC, human umbilical vein endothelial cell.

**Table 1 ijms-20-04406-t001:** More than a four-fold upregulation of DEGs in HUVECs transfected with miR-92a-3p.

No.	Gene Name	Symbol	RefSeq ID	Fold Change (miR-92a-3p/Control)
1	regulator of G-protein signaling 7	RGS7	NM_001282773.1	6.940
2	sterile alpha motif domain containing 15	SAMD15	NM_001010860.1	6.147
3	brain expressed X-linked 1	BEX1	NM_018476.3	6.144
4	Yip1 interacting factor homolog B, membrane trafficking protein	YIF1B	NM_001145461.1	5.812
5	IQ motif containing D	IQCD	XM_011537864.1	5.713
6	leukemia NUP98 fusion partner 1	LNP1	NM_001085451.1	5.566
7	phosphorylase kinase, alpha 2 (liver)	PHKA2	NM_000292.2	5.448
8	olfactory receptor family 2 subfamily T member 1	OR2T1	NM_030904.1	5.130
9	chromosome 15 open reading frame 48	C15orf48	NM_197955.2	5.092
10	kinesin family member 15	KIF15	NM_020242.2	4.765
11	dual adaptor of phosphotyrosine and 3-phosphoinositides	DAPP1	NM_001306151.1	4.626
12	DNA replication and sister chromatid cohesion 1	DSCC1	XM_005251065.2	4.596
13	proline/serine-rich coiled-coil 1	PSRC1	XM_011542306.1	4.579
14	glycerol-3-phosphate acyltransferase 2, mitochondrial	GPAT2	NM_207328.2	4.511
15	olfactory receptor, family 10, subfamily G, member 6	OR10G6	-	4.497
16	TYMS opposite strand	TYMSOS	NM_001012716.2	4.448
17	apolipoprotein O	APOO	NR_026545.2	4.444
18	membrane protein, palmitoylated 3	MPP3	XM_006721917.2	4.340
19	TNF alpha induced protein 8 like 2	TNFAIP8L2	NM_024575.4	4.338
20	WD repeat domain 63	WDR63	NM_001288563.1	4.328
21	pleckstrin	PLEK	NM_002664.2	4.324
22	solute carrier family 7 member 14	SLC7A14	NM_020949.2	4.306
23	MCM3AP antisense RNA 1	MCM3AP-AS1	NR_110565.1	4.220
24	zinc finger protein 114	ZNF114	NM_001301062.1	4.210
25	zinc finger RNA binding protein 2	ZFR2	NM_015174.1	4.188
26	neuralized E3 ubiquitin protein ligase 1	NEURL1	XM_011540334.1	4.143
27	mitogen-activated protein kinase 6 pseudogene 4	MAPK6PS4	-	4.132
28	chromosome 12 open reading frame 76	C12orf76	XM_005253882.2	4.039
29	zinc finger protein 485	ZNF485	XM_011539498.1	4.034
30	tripartite motif containing 45	TRIM45	XM_011542199.1	4.034

**Table 2 ijms-20-04406-t002:** More than a four-fold downregulation of DEGs in HUVECs transfected with miR-92a-3p.

No.	Gene Name	Symbol	RefSeq ID	Fold Change (miR-92a-3p/Control)
1	interleukin 1 receptor like 1	IL1RL1	NR_104167.1	0.076
2	interferon induced protein with tetratricopeptide repeats 2	IFIT2	NM_001547.4	0.076
3	thymidine phosphorylase	TYMP	NM_001257989.1	0.096
4	MX dynamin like GTPase 2	MX2	NM_002463.1	0.101
5	keratin 19, type I	KRT19	NM_002276.4	0.104
6	C-X-C motif chemokine ligand 10	CXCL10	NM_001565.3	0.117
7	C-X-C motif chemokine ligand 11	CXCL11	NM_001302123.1	0.118
8	radical S-adenosyl methionine domain containing 2	RSAD2	NM_080657.4	0.122
9	mannosidase alpha class 2A member 1	MAN2A1	NM_002372.3	0.133
10	interferon induced protein with tetratricopeptide repeats 1	IFIT1	NM_001548.4	0.133
11	cytidine/uridine monophosphate kinase 2	CMPK2	NR_046236.1	0.147
12	complement factor B	CFB	NM_001710.5	0.148
13	interferon stimulated exonuclease gene 20kDa	ISG20	XM_011521521.1	0.150
14	indoleamine 2,3-dioxygenase 1	IDO1	NM_002164.5	0.155
15	nidogen 2	NID2	XM_005267407.3	0.160
16	interferon, gamma-inducible protein 30	IFI30	NM_006332.4	0.166
17	C-C motif chemokine ligand 5	CCL5	NM_001278736.1	0.168
18	2′-5′-oligoadenylate synthetase-like	OASL	NM_001261825.1	0.169
19	dickkopf WNT signaling pathway inhibitor 3	DKK3	XM_006718178.2	0.170
20	general transcription factor IIE subunit 2	GTF2E2	XM_011544510.1	0.172
21	beta-2-microglobulin	B2M	NM_004048.2	0.174
22	matrix metallopeptidase 10	MMP10	NM_002425.2	0.174
23	prolyl 3-hydroxylase 3	P3H3	NM_014262.4	0.179
24	keratin 15, type I	KRT15	XM_005257345.2	0.181
25	interferon induced protein 35	IFI35	XM_005257302.3	0.191
26	basic leucine zipper ATF-like transcription factor 2	BATF2	XM_011544750.1	0.194
27	nuclear protein 1, transcriptional regulator	NUPR1	NM_012385.2	0.196
28	interferon induced protein with tetratricopeptide repeats 3	IFIT3	NM_001549.5	0.197
29	interferon induced transmembrane protein 1	IFITM1	NM_003641.3	0.198
30	solute carrier family 15 member 3	SLC15A3	XM_011545095.1	0.199
31	oleoyl-ACP hydrolase	OLAH	XM_006717456.2	0.200
32	mitogen-activated protein kinase-activated protein kinase 2	MAPKAPK2	NM_032960.3	0.201
33	ISG15 ubiquitin-like modifier	ISG15	NM_005101.3	0.203
34	AXL receptor tyrosine kinase	AXL	NM_001278599.1	0.203
35	atypical chemokine receptor 3	ACKR3	NM_020311.2	0.207
36	complement component 3a receptor 1	C3AR1	NM_004054.2	0.207
37	MX dynamin like GTPase 1	MX1	NM_001144925.2	0.208
38	tumor necrosis factor superfamily member 13b	TNFSF13B	XM_005254029.3	0.208
39	isocitrate dehydrogenase 1 (NADP+)	IDH1	NM_005896.3	0.208
40	PDZ domain containing 2	PDZD2	NM_178140.3	0.210
41	integrin subunit alpha 5	ITGA5	NM_002205.3	0.214
42	tumor-associated calcium signal transducer 2	TACSTD2	NM_002353.2	0.219
43	HLA complex P5 (non-protein coding)	HCP5	NR_040662.1	0.221
44	olfactory receptor family 9 subfamily I member 1	OR9I1	NM_001005211.1	0.222
45	fibrillin 1	FBN1	NM_000138.4	0.222
46	phospholipase A1 member A	PLA1A	NM_001293225.1	0.223
47	CD69 molecule	CD69	NM_001781.2	0.224
48	integral membrane protein 2B	ITM2B	NM_021999.4	0.226
49	DnaJ heat shock protein family (Hsp40) member B9	DNAJB9	NM_012328.2	0.228
50	scavenger receptor class B member 2	SCARB2	NM_001204255.1	0.230
51	sterile alpha motif domain containing 9	SAMD9	NM_017654.3	0.230
52	insulin like growth factor binding protein 6	IGFBP6	NM_002178.2	0.231
53	claudin 11	CLDN11	NM_001185056.1	0.232
54	unc-93 homolog B1 (C. elegans)	UNC93B1	XM_011545291.1	0.238
55	FRY microtubule binding protein	FRY	XM_006719749.2	0.239
56	myosin VIIA and Rab interacting protein	MYRIP	NR_104316.1	0.240
57	selectin E	SELE	NM_000450.2	0.240
58	interferon regulatory factor 7	IRF7	XM_005252909.2	0.244
59	HECT and RLD domain containing E3 ubiquitin protein ligase 5	HERC5	NM_016323.3	0.244
60	secreted and transmembrane 1	SECTM1	XM_011523588.1	0.244
61	laminin subunit gamma 2	LAMC2	NM_005562.2	0.248

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
