# Peer review of "Extracellular Vesicles Containing MicroRNA-92a-3p Facilitate Partial Endothelial-Mesenchymal Transition and Angiogenesis in Endothelial Cells"

_ijms, 2019, doi:10.3390/ijms20184406_

Round 1
Reviewer 1 Report
miRNAs are short non-coding RNAs that can dampen gene expression by binding to mRNA targets in a sequence-specific manner. Aberrant expression of mature miRNAs of the miR-17-92 cluster, among them miR-92a-3p was repeatedly observed in human tumors of various origin, and their presence in extracellular membrane vesicles has been reported by different groups for several tumor entities, including colorectal cancer. Extracellular membrane vesicles may mediate the intercellular transfer of proteins or non-coding RNAs, however, the physiological relevance of microvesicle-derived miRNAs (and proteins) is far from being settled.
In the current manuscript, Yamada and colleagues extend previous work from their lab on the function of microvesicle-derived miR-92a-3p.
As their miR-92a-3p-excreting model system the authors work with the human colon adenocarcinoma DLD-1 cell line. They show that these cells express miR-92a-3p and that they secrete vesicles that contain miR-92a-3p. Next, they exposed fluorescently labeled vesicles to endothelial HUVEC cells and confirmed vesicular uptake and a moderate increase of miR-92a-3p levels in the HUVEC cells. They associate the uptake with an increase in proliferation, migration and tube formation (an in vitro angiogenesis assay). Hence, they conclude that miR-92a-3p promotes angiogenesis.
Subsequently, the authors transfected HUVEC cells with miR-92a-3p and identified >1000 differentially expressed genes as compared to the control cells, among them CLDN11 – a component of tight junctions with functions in proliferation and migration. Next, Yamada and colleagues showed that ectopic expression of miR-92a-3p in HUVEC cells reduces CLDN11 mRNA and protein levels, and that the CLDN11 3’UTR indeed harbors one direct binding site for miR-92a-3p. Knockdown of CLDN11 reproduced the proliferation, migration and angiogenesis phenotypes induced by extracellular vesicles or ectopic miR-92a-3p expression, albeit it was more potent.
Finally, the authors hypothesize that miR-92a-3p directs a network of genes involved in endothelial-to-mesenchymal transition, promoting tumor vascularization.
Overall, the manuscript is well written and the materials and methods section contains all necessary information. The identification of CLDN11 is to my knowledge novel and the most convincing part of the manuscript (Fig. 3 and Fig. 4). The initial part on microvesicle-derived miR-92a-3p (Fig. 1 and Fig. 2) is interesting and in line with current intense research efforts on microvesicle-derived cargo and its function in recipient cells. This initial part still requires some additional work to show that extravesicular miR-92a-3p can really do the job in the HUVEC cells that take these vesicles up.
Specifically, my worries are that although the experiments shown in Fig. 3 and Fig. 4 are quite straight-forward, they rely on the ectopic expression of miR-92a-3p that yields substantially higher intracellular levels of the miRNA (appr. 16-fold according to Fig. 2E) as compared to the levels achieved by vesicular uptake (appr. 0.25-fold according to Fig. 2C).
Major points:
The increase of miR-92a-3p upon treatment of HUVEC cells with fluorescently labeled DLD-1-derived vesicles is rather minor (Fig. 2C), particularly when compared to the miR-92a-3p overexpression experiment (Fig. 2E). Given that miRNAs mostly fine tune mRNA expression it is rather unlikely that such as minor increase in expression efficiently promotes mRNA repression. The authors critically need to show more convincing data here. One option to strengthen their point would be that they determine expression levels of other miR-92 family members such as but not restricted to miR-25-3p (in Fig. 2C) that have a largely overlapping set of mRNA targets.
Also, at 48 hours post vesicle-treatment, are there still fluorescently labeled vesicles in the HUVEC cells and are they still intact? I am not a specialist on extracellular vesicles, but in order to be effective the miR-92a-3p miRNAs have to be released into the cytoplasm of the HUVEC cells and I wonder if this release had already happened 48 hours post treatment. I understand that this may be a tricky question, but can the authors perform a simple experiment here or at least comment on this issue in their results or discussion part?
It is completely unclear to me how cell viability was determined and normalized for Fig. 2B and D as well as Fig. 4E. How can viability exceed 100%? Could it be that the authors measure here rather cell proliferation than cell viability (as indicated for Fig. 4E in the text)? This has to be clarified.
In Figure 5A, where are the consequently proposed target genes (Snail, Vimentin, ZO-1) located? Even if the authors put this figure forward as a hypothesis, it is not very convincing and I am not sure that it really helps the manuscript as it opens a new unexplored box rather than closing one.
For Western blots (Fig. 4C and Fig. 5B), in addition to the representative picture a graph showing at least n=3 for the depicted data points is necessary.
Author Response
Responses to Reviewer #1
We thank Reviewer #1 for their valuable comments. We have addressed all these comments in our revised manuscript as follows:
miRNAs are short non-coding RNAs that can dampen gene expression by binding to mRNA targets in a sequence-specific manner. Aberrant expression of mature miRNAs of the miR-17-92 cluster, among them miR-92a-3p was repeatedly observed in human tumors of various origin, and their presence in extracellular membrane vesicles has been reported by different groups for several tumor entities, including colorectal cancer. Extracellular membrane vesicles may mediate the intercellular transfer of proteins or non-coding RNAs, however, the physiological relevance of microvesicle-derived miRNAs (and proteins) is far from being settled.
In the current manuscript, Yamada and colleagues extend previous work from their lab on the function of microvesicle-derived miR-92a-3p.
As their miR-92a-3p-excreting model system the authors work with the human colon adenocarcinoma DLD-1 cell line. They show that these cells express miR-92a-3p and that they secrete vesicles that contain miR-92a-3p. Next, they exposed fluorescently labeled vesicles to endothelial HUVEC cells and confirmed vesicular uptake and a moderate increase of miR-92a-3p levels in the HUVEC cells. They associate the uptake with an increase in proliferation, migration and tube formation (an in vitro angiogenesis assay). Hence, they conclude that miR-92a-3p promotes angiogenesis.
Subsequently, the authors transfected HUVEC cells with miR-92a-3p and identified >1000 differentially expressed genes as compared to the control cells, among them CLDN11 – a component of tight junctions with functions in proliferation and migration. Next, Yamada and colleagues showed that ectopic expression of miR-92a-3p in HUVEC cells reduces CLDN11 mRNA and protein levels, and that the CLDN11 3’UTR indeed harbors one direct binding site for miR-92a-3p. Knockdown of CLDN11 reproduced the proliferation, migration and angiogenesis phenotypes induced by extracellular vesicles or ectopic miR-92a-3p expression, albeit it was more potent.
Finally, the authors hypothesize that miR-92a-3p directs a network of genes involved in endothelial-to-mesenchymal transition, promoting tumor vascularization.
Overall, the manuscript is well written and the materials and methods section contains all necessary information. The identification of CLDN11 is to my knowledge novel and the most convincing part of the manuscript (Fig. 3 and Fig. 4). The initial part on microvesicle-derived miR-92a-3p (Fig. 1 and Fig. 2) is interesting and in line with current intense research efforts on microvesicle-derived cargo and its function in recipient cells. This initial part still requires some additional work to show that extravesicular miR-92a-3p can really do the job in the HUVEC cells that take these vesicles up.
Specifically, my worries are that although the experiments shown in Fig. 3 and Fig. 4 are quite straight-forward, they rely on the ectopic expression of miR-92a-3p that yields substantially higher intracellular levels of the miRNA (appr. 16-fold according to Fig. 2E) as compared to the levels achieved by vesicular uptake (appr. 0.25-fold according to Fig. 2C).
Response: We have added additional experiments showing not only ectopic expression of miR-92a-3p but also that EV uptake can mediate partial EndoMT. We also now show that the intracellular levels of miR-92a-3p achieved via EV uptake are sufficient to modulate intracellular signaling (Figs. 2, 3, and 4).
Major points:
The increase of miR-92a-3p upon treatment of HUVEC cells with fluorescently labeled DLD-1-derived vesicles is rather minor (Fig. 2C), particularly when compared to the miR-92a-3p overexpression experiment (Fig. 2E). Given that miRNAs mostly fine tune mRNA expression it is rather unlikely that such as minor increase in expression efficiently promotes mRNA repression. The authors critically need to show more convincing data here. One option to strengthen their point would be that they determine expression levels of other miR-92 family members such as but not restricted to miR-25-3p (in Fig. 2C) that have a largely overlapping set of mRNA targets.
Response: We have added additional experiments showing that the increase in miR-92a-3p expression achieved via EV uptake is sufficient to inhibit mRNA expression (Figs. 2, 3, and 4).
Also, at 48 hours post vesicle-treatment, are there still fluorescently labeled vesicles in the HUVEC cells and are they still intact? I am not a specialist on extracellular vesicles, but in order to be effective the miR-92a-3p miRNAs have to be released into the cytoplasm of the HUVEC cells and I wonder if this release had already happened 48 hours post treatment. I understand that this may be a tricky question, but can the authors perform a simple experiment here or at least comment on this issue in their results or discussion part?
Response: Previous data from our laboratory together with results presented in this manuscript suggest that miR-92a-3p has already been released from EVs at 24 h post treatment. Fluorescently-labeled EVs gradually increased in HUVECs until 16 h post treatment (Fig 2A) and then gradually decrease, which is consistent with our previous study published in Biochimica et Biophysica Acta in 2014. We have addressed this issue by adding text to the Results section of our revised manuscript.
It is completely unclear to me how cell viability was determined and normalized for Fig. 2B and D as well as Fig. 4E. How can viability exceed 100%? Could it be that the authors measure here rather cell proliferation than cell viability (as indicated for Fig. 4E in the text)? This has to be clarified.
Response: We thank the reviewer for pointing out our mistake. We did indeed measure cell proliferation using the Trypan blue exclusion test and not cell viability. We have replaced “cell viability (%)” with “cell proliferation ratio (%)” in Figs. 2B, 2D, and 4E.
In Figure 5A, where are the consequently proposed target genes (Snail, Vimentin, ZO-1) located? Even if the authors put this figure forward as a hypothesis, it is not very convincing and I am not sure that it really helps the manuscript as it opens a new unexplored box rather than closing one.
Response: Data for the proposed targets snail and vimentin are not presented in Fig. 5A but are shown in Figs. 5C and/or 5D. Snail is a transcriptional factor related to EMT induction and vimentin upregulation is a marker of EMT induction. Data for ZO-1 are presented in Fig. 5A as TJP1 as well as in Fig. 5C. In Figs. 1 through 4, we provide data showing that EVs containing miR-92a-3p induce an EMT-like phenomenon in endothelial cells; in Fig. 5, we show evidence that an EMT-like process is occurring and propose that this process be called partial EndoMT.
For Western blots (Fig. 4C and Fig. 5B), in addition to the representative picture a graph showing at least n=3 for the depicted data points is necessary.
Response: We have inserted graphs showing densitometric values (n=3) calculated using Image J for each immunoblot.
Reviewer 2 Report
Extracellular vesicles containing microRNA-92a-3p facilitate partial endothelial mesenchymal transition and angiogenesis in endothelial cells
Nami O. Yamada 1,*, Kazuki Heishima 2, Yukihiro Akao 2 and Takao Senda 1
Comments for the Authors
The manuscript entitled ‘Extracellular vesicles containing microRNA-92a-3p facilitate partial endothelial mesenchymal transition and angiogenesis in endothelial cells’ by Yamada NO and co-workers describes the molecular mechanism of EV carried microRNA-92a-3p induced partial Endothelial-Mesenchymal transition and angiogenesis. After ectopic expression of miR-92a-3p in endothelial cells to identify novel miR-92a-3p targets. Intriguingly, claudin-11 identified as a novel target of miR-92a-3p. Therefore, it indicates that EV/miR-92a-3p target Claudin-11 to induce partial endothelial-to-mesenchymal transition in endothelial cells.
Major comments
In Figure 1, panel C, it is very hard to identify the 100 nm and 200 nm, etc. with low resolution. Based on NTA result (Figure 1C), Authors stated that “As shown in Fig. 1C, nanoparticle tracking analysis (NTA) revealed that DLD-1 cells-derived EVs were heterogeneous in size (average size: 82.8 ± 36.4 nm), indicating that the EVs secreted from DLD-1 cells are a mixture of exosomes (30-100 nm) and shed-microvesicles (100-400 nm).” It is not appropriate to say it is a mixture of exosomes and shed microvesicles, and it requires more evidence to support it.
1) EV preparations not only contain Exos markers (Fig 1D) but also contain sMVs markers.
2) TEM or CryoEM is required to show the morphology of EVs prepared from 40 mL CM (it is recommended by MISEV2018, PMID: 30637094 ).
3) Based on the literature, the size of exosomes can be up to 150 nm.
Authors mentioned normal colon In Figure 1 A and Figure 4A, while there is no detailed description of the normal colon? Where are ethics?
Could the authors explain why PBS treated HUVECs has around “6” miR-92a-3 relative expression/RNU6B in Figure 2C, while in Figure 2E, there is only less than “2” miR-92a-3 relative expression/RNU6B
To finally get the conclusion that EV derived miR-92a-3p downregulate Claudin-11 expression to mediate partial EMT in endothelial cells, it required several critical experiments:
1) like in Figure 2 F/G, please demonstrate purified EVs (contain miR-92a-3p) impact on migration and angiogenesis of HUVECs,
2) Similar like in Figure 4 D)purified EVs (contain miR-92a-3p) impact in Human Claudin-11 luciferase assay (3`UTR of CLDN11 mRNA,
3) Purified EVs (contain miR-92a-3p) induces “Partial ” EndoMT
4) whether inhibitors (GW4869) treated colon cancer cells secreted EVs will impair EV-miR-92a-3p mediate EndoEMT
Otherwise, it just demonstrates miR-92a-3p mediated EndoMT on HUVECs rather than EV derived miR-92a-3p mediated EndoMT on HUVECs
Proliferation assay was missing in the M&M; please give detailed information such as how many EV incubated with HUVECs
Minor comments
It is not very clear what condition treated HUVECs used in cDNA microarray? 10 nm miR92a-3p? how many hours post-transfection? As mentioned In Figure 2, the phenomena confirmed under 10 nm miR-92a-3p, 24hour post-transfection condition.
In Figure 3B, it is impossible to read the different Gene Ontology with low resolution.
How many EV proteins isolation from 40 ml Condition medium from 1*10^6 cells?
In the 4.2 P12, “rpm” was mentioned, please indicate which equipment and rotor used in EV isolation, please be aware the difference between rpm and rcf.
Author Response
Responses to Reviewer #2
We would also like to thank Reviewer #2 for their valuable suggestions. We have addressed all the reviewers’ concerns in our revised manuscript as follows:
The manuscript entitled ‘Extracellular vesicles containing microRNA-92a-3p facilitate partial endothelial mesenchymal transition and angiogenesis in endothelial cells’ by Yamada NO and co-workers describes the molecular mechanism of EV carried microRNA-92a-3p induced partial Endothelial-Mesenchymal transition and angiogenesis. After ectopic expression of miR-92a-3p in endothelial cells to identify novel miR-92a-3p targets. Intriguingly, claudin-11 identified as a novel target of miR-92a-3p. Therefore, it indicates that EV/miR-92a-3p target Claudin-11 to induce partial endothelial-tomesenchymal transition in endothelial cells.
Major comments
In Figure 1, panel C, it is very hard to identify the 100 nm and 200 nm, etc. with low resolution. Based on NTA result (Figure 1C), Authors stated that “As shown in Fig. 1C, nanoparticle tracking analysis (NTA) revealed that DLD-1 cells-derived EVs were heterogeneous in size (average size: 82.8 ± 36.4 nm), indicating that the EVs secreted from DLD-1 cells are a mixture of exosomes (30-100 nm) and shed-microvesicles (100- 400 nm).” It is not appropriate to say it is a mixture of exosomes and shed microvesicles, and it requires more evidence to support it.
Response: We replaced Fig. 1C with a scatter plot graph created in Excel based on the raw NTA data. To address Reviewer #2’s suggestion, we add more characterization of our EVs and concluded that the EVs collected from DLD-1 cells are mainly exosomes.
EV preparations not only contain Exos markers (Fig 1D) but also contain sMVs markers.
Response: We now present data from analyzing additional markers (i.e. TSG101, flotillin-1, and actinin-4) to distinguish exosomes from sMVs (Fig. 1D). TSG101 was dominantly expressed, but the plasma membrane-related protein flotillin-1 and the cytoskeletal protein actinin-4 were barely detected in our EVs.
1) TEM or CryoEM is required to show the morphology of EVs prepared from 40 mL CM (it is recommended by MISEV2018, PMID: 30637094 ).
Response: We have added data from TEM experiments (Fig. 1E).
3) Based on the literature, the size of exosomes can be up to 150 nm. Response: We replaced the exosome size range of 30-100 nm with 30-150 nm.
Authors mentioned normal colon In Figure 1 A and Figure 4A, while there is no detailed description of the normal colon? Where are ethics?
Response: Total RNA from normal human colon tissue was purchased from Clontech (Mountain View, CA). We now state this information in the Materials and Methods section of our revised manuscript.
Could the authors explain why PBS treated HUVECs has around “6” miR-92a-3 relative expression/RNU6B in Figure 2C, while in Figure 2E, there is only less than “2” miR-92a-3 relative expression/RNU6B
Response: In Fig. 2E, we converted the relative expression level of miR-92a-3p/RNU6B in PBS-treated HUVECs to 1, but we did not present the expression data in Fig. 2C in this same way. We apologize for the confusion. We have changed how the expression data are presented in Fig. 2C of our revised manuscript to be consistent with how relative expression levels are presented in Fig. 2E.
To finally get the conclusion that EV derived miR-92a-3p downregulate Claudin-11 expression to mediate partial EMT in endothelial cells, it required several critical experiments:
1) like in Figure 2 F/G, please demonstrate purified EVs (contain miR-92a-3p) impact on migration and angiogenesis of HUVECs,
Response: We now present experimental data showing that EVs increase migration and tube formation activity of HUVECs (Figs. 2F and 2G).
2) Similar like in Figure 4 D)purified EVs (contain miR-92a-3p) impact in Human Claudin-11 luciferase assay (3`UTR of CLDN11 mRNA,
Response: We have added data from luciferase assays using EVs and reporter vectors (3’UTR of CLDN11 mRNA) (Fig. 4D).
3) Purified EVs (contain miR-92a-3p) induces “Partial ” EndoMT
Response: We now include data showing that EVs can induce partial EndoMT (Fig. 5).
4) whether inhibitors (GW4869) treated colon cancer cells secreted EVs will impair EVmiR-92a-3p mediate EndoEMT
Otherwise, it just demonstrates miR-92a-3p mediated EndoMT on HUVECs rather than EV derived miR-92a-3p mediated EndoMT on HUVECs
Response: We agree with the reviewer that data from this important experiment would strengthen our conclusions. We actually previously tried using the nSMase2 inhibitor to stop miR-92a secretion in DLD-1 cells. However, we could not achieve inhibition. We talked with the authors of the article “Neutral sphingomyelinase 2 (nSMase2)-dependent exosomal transfer of angiogenic microRNAs regulate cancer cell metastasis” published in the Journal of Biological Chemistry in 2013 at a cancer conference; they said they could not always achieve an inhibitory effect on EV secretion depending on the circumstances. We addressed Reviewer #2’s concerns by performing an alternative experiment; we now show new data demonstrating that EVs derived from antagomiR-92a-3p (miR-92a-3p inhibitor)-transfected colon cancer cells impaired EV-mediated EndoMT (Fig. 5).
Proliferation assay was missing in the M&M; please give detailed information such as how many EV incubated with HUVECs
Response: We have added information regarding the Trypan blue exclusion test proliferation assay and how many EVs were incubated with HUVECs to the Materials and Methods section of our revised manuscript.
Minor comments
It is not very clear what condition treated HUVECs used in cDNA microarray? 10 nm miR92a-3p? how many hours post-transfection? As mentioned In Figure 2, the phenomena confirmed under 10 nm miR-92a-3p, 24hour post-transfection condition.
Response: We have added the requested information in section 4.7 of our revised manuscript.
In Figure 3B, it is impossible to read the different Gene Ontology with low resolution.
Response: We replaced Fig. 3B with a higher resolution one.
How many EV proteins isolation from 40 ml Condition medium from 1*10^6 cells?
Response: We can extract about 100-200 μg of EV proteins from 40 mL of medium from 1*10^6 cells.
In the 4.2 P12, “rpm” was mentioned, please indicate which equipment and rotor used in EV isolation, please be aware the difference between rpm and rcf.
Response: We have added information regarding the equipment and rotor used for EV isolation in section 4.2 of our revised manuscript.
Round 2
Reviewer 2 Report
In Figure 1 E, TEM results did not show typical cup-shape morphology of exosomes. It might be the the difference between phosphotungstic acid staining and uranyl acetate staining in TEM protocol <PMID:30651939>. If you want to improve the quality of morphology of exosomes, uranyl acetate based TEM or if possible, CryoEM is a good option.